Effects of virtual reality exercise on fatigue, pain, and psychological status among cancer patients: a meta-analysis

Wang Ruihan 1
Wang Yunwei 1
Li Shiming 2 haiyanglishiming@163.com
Li Shuoqi 1 3 lishuoqi@ntu.edu.cn
1 School of Sports Science, Nantong University , Nantong , China
2 Department of Physical Education, Ocean University of China , Qingdao , China
3 Rehabilitation Medicine Department, Nantong Key Laboratory of Sports and Exercise Rehabilitation Science , Nantong , China
Epifania Ottavia
Electronic publication date: 2025 Oct 29
Publication date: 2025
Volume: 13
Electronic Location ID: e20196
Received 2025 May 19; Accepted 2025 Sep 15
Copyright: © 2025 Wang et al.
Copyright year: 2025
Copyright holder: Wang et al.
License: This is an open access article distributed under the terms of the Creative Commons Attribution License, which permits unrestricted use, distribution, reproduction and adaptation in any medium and for any purpose provided that it is properly attributed. For attribution, the original author(s), title, publication source (PeerJ) and either DOI or URL of the article must be cited.
License URL: https://creativecommons.org/licenses/by/4.0/

Keywords: Cancer, Virtual reality, Exercise, Fatigue, Pain

Funding: Shandong Provincial Natural Science Foundation of China ZR2022MC205 Research Project on Undergraduate Teaching Reform in Shandong Province Z2024065 Nantong University Doctoral Initiation Fund 135423619048 The project supported by the Shandong Provincial Natural Science Foundation of China (No. ZR2022MC205), the Research Project on Undergraduate Teaching Reform in Shandong Province (Key project, No. Z2024065), and the Nantong University Doctoral Initiation Fund (No. 135423619048). The funders had no role in study design, data collection and analysis, decision to publish, or preparation of the manuscript.

==============================
Background

Cancer and its associated therapies can result in pain, fatigue, and high psychological depression with anxiety levels. Therefore, this meta-analysis explored the effects of virtual reality (VR) exercise on fatigue and pain in cancer patients, and the secondary outcomes assessed the levels of depression and anxiety.

Methodology

Eligible studies were searched for through four databases and then screened. The inclusion criteria are as follows: (1) Participants with cancer; (2) randomised controlled trial and single-arm trials; (3) the intervention group received VR exercise; (4) conducted pre- and post-test, which include fatigue, pain, depression and anxiety. The Cochrane bias risk assessment tool was used to evaluate the quality of the selected study. Standardized Mean Difference (SMD) was selected as the appropriate effect scale index, and Revman 5.4 software was used to analyze the mean difference of the selected article data. (Registration number: CRD420251037421).

Results

The meta-analysis outcomes implied a greater reduction in fatigue following the virtual reality group (VRG) intervention in comparison to control group (COG) (SMD, 0.77 [0.50, 1.04], p < 0.05, I2 = 61%). The heterogeneity of these outcomes then indicated that subgroup analyses were necessary. Consequently, these analyses denoted that fatigue was substantially improved in intervention durations below 30 min (SMD, 0.44 [0.08, 0.81], p < 0.05, I2 = 22%) and above 30 min (SMD, 1.20 [0.78, 1.61], p < 0.05, I2 = 0%). Significant pain improvement was also observed in VRG than COG (SMD, −0.88 [−1.15, −0.60], p < 0.05, I2 = 36%).

Conclusions

This review indicated that VR exercise reduced fatigue, pain, and anxiety in cancer patients. Nevertheless, the effect of VR exercise on relieving depression is not clear, which may be a potentially effective non pharmacological intervention in this population.

Introduction

Cancer is a significant healthcare concern, with 19.3 million new cases reported worldwide in 2020 and an anticipated 28.4 million new cases by 2040 (Sung et al., 2021). Various cancer treatment options are also frequently employed, such as chemotherapy, hormone therapy, radiation therapy, and surgery (Alvarado-Omenat et al., 2025). Nonetheless, cancer and its associated therapies can result in pain, fatigue, and high psychological depression with anxiety levels (McConnell, Scott & Porter, 2016). These adverse effects can lead to a reduction in overall survival rates by diminishing treatment adherence. Thus, examining non-pharmacological intervention strategies with reduced adverse effects has emerged as a pressing issue requiring resolution.

Numerous evidence-based studies have demonstrated the advantages of exercise rehabilitation in mitigating the negative effects of cancer. These benefits encompass enhanced muscle power, physical function, and psychological well-being (Fors et al., 2011; Wirtz & Baumann, 2018). Hence, virtual reality (VR) has been progressively integrated into rehabilitation practices to improve the advantages of physical rehabilitation (Chan & Ismail, 2014). This technology is a type of artificial intelligence (AI) that was first established in the 1960s and has undergone gradual development throughout the 20th century (Cipresso et al., 2018). Current technological advancements have also rendered AI-related applications in the medical sector increasingly accessible (Ayed et al., 2019; Karamians et al., 2020). One prominent example is users engaging multiple senses (touch, hearing, vision) in physical rehabilitation utilising VR through the integration of computerised systems and multidimensional graphics (Li et al., 2017). This process significantly enhances motivation for patients participating in therapeutic interventions (Zhang et al., 2022). Alternatively, integrating VR with a game can create an interactive environment for the patient (Vernadakis et al., 2014).

A previous meta-analysis (Zhang et al., 2022) exploring the impact of VR intervention on breast cancer showed that VR intervention could improve fatigue and upper limb abduction function. However, few studies were included, and at most three studies were included in each parameter, which restricted the reliability of the results. In addition, it did not assess pain and depression levels. The efficacy of exercise and VR interventions concerning pain and fatigue in cancer patients has remained uncertain. The studies have indicated that the combination of exercise and VR may be more efficient in alleviating pain and fatigue in cancer patients than exercise rehabilitation alone (Abdelmoniem Ibrahim et al., 2024; da Silva Alves et al., 2023). Conversely, Hamari et al. (2019) documented an insignificant difference between the two groups. Consequently, this meta-analysis explored the effects of VR exercise on fatigue and pain in cancer patients, with secondary outcomes assessing depression and anxiety levels.

Survey methodology

Protocol and registration

This review was preregistered within the International Prospective Register of Systematic Reviews (CRD420251037421). Considering that each data point employed in this analysis was sourced, ethical approval and informed consent were deemed unnecessary.

Data sources and study selection

Relevant journal articles published between 1st January 2015 and 18th April 2025 were determined through a literature search across four electronic databases: (i) EBSCO, (ii) Scopus, (iii) PubMed, and (iv) Web of Science. This review utilised several subject headings and keywords, including “Virtual Reality”, “Exercise”, “Training”, and “Cancer”. The reference lists of all included articles and recent reviews were manually screened to identify other eligible studies. Specific information regarding this search strategy is provided in Materials S1. A manual review of the reference lists and pertinent reviews was also conducted to identify and acquire additional relevant studies. Two researchers (R.W and Y.W) independently conducted database searches. If there is a disagreement, it will be resolved by the third researcher (S.L). The flow diagram of search results is shown in Fig. 1.

Figure 1 Flow diagram of the search results using the preferred reporting items for systematic reviews and meta-analysis (PRISMA).

*Consider, if feasible to do so, reporting the number of records identified from each database or register searched (rather than the total number across all databases/registers). **If automation tools were used, indicate how many records were excluded by a human and how many were excluded by automation tools.

Inclusion and exclusion criteria

This review employed multiple inclusion criteria as follows: i. To comprehensively evaluate the impact of VR intervention on cancer patients, randomized controlled trials (RCT) and single-arm trials were included (Chen et al., 2024).

ii. Participants diagnosed with cancer.

iii. Exercise-based articles using VR as the experimental group (VRG).

iv. The RCT must involve usual care as a control group (COG).

v. Measurement of depression, anxiety, pain, and fatigue before and after interventions.

vi. Reporting of test indicators as mean (standard deviation) or median (interquartile range).

vii. English language-based full-text articles. This review did not include abstracts, poster presentations,gray literature, and conference proceedings.

Quality assessment

This review applied a quality assessment methodology based on the authors’ previous study (Li et al., 2022). The method involved utilising the Physical Therapy Evidence Database Scale (PEDro) with 11 questions to evaluate the quality of selected RCT articles (de Morton, 2009). Each PEDro item was scored as “1” (met) or “0” (not met), with a total score ranging from 0 to 10. A score ≥6 was defined as high quality, reflecting rigorous methodology (e.g., randomization, allocation concealment, and blinding) (Suárez-Iglesias et al., 2019). The Quality Assessment Tool for Before-After (Pre-Post) Studies with No Control Group (NIH, 12 questions) scale was also utilised to assess the quality of the included single-arm trials (Ma et al., 2020). The NIH tool categorizes studies into three levels: “poor” (≤40% of items met), “fair” (41–70% met), and “good” (≥71% met). Likewise, the Levels of Evidence for Therapeutic Studies tool (containing 10 levels) was employed to compute the evidence levels (Burns, Rohrich & Chung, 2011). The Grade Practice Recommendations tool was also used to examine the recommendations levels, which the American Society of Plastic Surgeons developed. This tool was outlined in the evidence-based clinical practice framework (Burns, Rohrich & Chung, 2011). For discrepancies in PEDro and NIH scoring, two independent reviewers (R.W and Y.W) initially evaluated each article; disagreements were resolved through discussion with a third reviewer (S.L) to ensure consistency.

Data extraction

An independent screening of the titles and abstracts was performed by two researchers (R.W and Y.W). This process was followed by a review of the complete texts according to the inclusion criteria, with verification by a third researcher (S.L). Disagreements at this stage were addressed through discussion with an additional reviewer to reach a final decision. Data were then manually extracted by two independent reviewers for studies meeting the inclusion criteria, utilising predesigned structured extraction forms to record duration, gender, index, age, disease, and VR programme. The extraction of VR exercise intervention program mainly includes VR equipment, intervention methods, repetition times, and intervention time. For fatigue and pain outcomes, data before and after interventions were extracted for both the VRG and COG. For anxiety and depression outcomes, data were extracted only for the VRG before and after the intervention.

Sensitivity analysis

The methods of sensitivity analysis mainly include single study exclusion analysis and statistical model sensitivity test.

Statistical analyses

All analyses were conducted using Review Manager (version 5.4.1; The Cochrane Collaboration, Copenhagen, Denmark). The data were analysed based on the information available in the article in cases where the article did not present an effect size of interest for each outcome. This process followed the methodology outlined in the Cochrane Handbook for Systematic Reviews of Interventions (Zhang et al., 2024). Even though the findings of the included articles were continuous variables, the testing techniques varied. Thus, the effect scale index was represented by the standardised mean difference (SMD). Hozo, Djulbegovic & Hozo’s (2005) methodology of converting median (range) to mean (standard deviation) was also used if the outcomes in the articles contained median (range) (Principles of Data Conversion: http://vassarstats.net/median_range.html).

The I2 statistic was applied to assess the heterogeneity among the studies, in which a lower heterogeneity among the studies was denoted if I2 was reduced. No heterogeneity among the studies was observed if I2 was below 50%. This outcome implied that the analysis necessitated a fixed effect model. On the contrary, an I2 equal to or higher than 50% indicated heterogeneity among the studies. This finding signified that the analysis required a random effect model (Li et al., 2021). Funnel and forest plots were also employed to investigate publication bias and SMD, respectively. The uncertainty was quantified using 95% confidence intervals (95% CI). In addition, the Egger’s test was used to test the publication bias, and a p value greater than 0.05 represents no significant publication bias.

Results

Eligibility of articles

This review discovered that the relationships between VR exercise, depression, anxiety, fatigue, and pain among cancer patients were explored in 12 articles (RCT articles = 8 and non-RCT articles = 4) (see Table 1). The comparative analysis involving VRG and COG incorporated fatigue and pain evaluations. Meanwhile, depression and anxiety assessment were included solely in the baseline and post-intervention data. These articles also received ethical approval from their respective institutions, and the two researchers achieved a Cohen’s kappa coefficient of 0.898. The 12 articles comprised 108 males and 254 females, with five articles (Abdelmoniem Ibrahim et al., 2024; da Silva Alves et al., 2023; Hamari et al., 2019; da Silva Alves et al., 2017; Villumsen et al., 2019) reporting fatigue and four articles documenting pain (Abdelmoniem Ibrahim et al., 2024; Park et al., 2023; Feyzioğlu et al., 2020; Basha et al., 2022), anxiety (Abdelmoniem Ibrahim et al., 2024; Villumsen et al., 2019; Mao, Chen & Wang, 2024; Qi et al., 2024) and depression (Mao, Chen & Wang, 2024; Qi et al., 2024; Tsuda et al., 2016; House et al., 2016). Moreover, the intervention period ranged from a minimum of 3 weeks to a maximum of 12 months.

Table 1 Characteristics of included studies.

Study	Age (y)	Gender	Disease	Duration	VR program	Index	
da Silva Alves et al. (2017)	57.1 ± 16.7	1M/14F	Cancer	10 weeks; 2x/week	Xbox 360 Kinect; Physical games aimed at mobilizing the upper and lower limbs; 20 sets; 47 min	Fatigue	
da Silva Alves et al. (2023)	60.1 ± 12.1	14M/24F	Cancer	7 weeks; 3x/week	Xbox 360 Kinect; Physical games aimed at mobilizing the upper and lower limbs; 20 sets; 47 min	Fatigue	
Basha et al. (2022)	48.8 ± 7.0	30F	Breast cancer	8 weeks; 5x/week	Xbox 360 Kinect; VR Kinect; Physical games; 40 sets	Pain	
Feyzioğlu et al. (2020)	50.84 ± 8.53	20F	Breast cancer	6 weeks; 2x/week	Xbox 360 Kinect; Kinect Sports game: darts, bowling, boxing, beach volleyball, table tennis and Fruit Ninja; 12 sets; 45 min	Pain	
Hamari et al. (2019)	7.8 ± 3.3	12M/5F	Cancer	8 weeks; 7x/week	Nintendo Wii Fit; Balance-oriented virtual game campaigns; 30 min	Fatigue	
House et al. (2016)	57.8 ± 20.4	6F	Breast cancer	8 weeks; 2x/week	Bright Arm Duo Rehabilitation System; Card Island and Remember that Card, Musical Drums, Xylophone game, Pick & Place; 20–50 min	Pain; Depression	
Abdelmoniem Ibrahim et al. (2024)	47 ± 3.9	21F	Cancer	8 weeks; 3x/week	Pablo games; Recycle, Firefighters, Shooting Cans, Balloon, and Apple Hunter; 24 sets; 15 min	Fatigue; Pain; Anxiety	
Mao, Chen & Wang (2024)	43.6 ± 3.3	48F	Cancer	10 weeks; 1x/week	Pico Neo 4; personalized curriculum, intelligent monitoring, emotion tracking, and Funny Games; 10 sets; 60 min	Anxiety; Depression	
Park et al. (2023)	42.6 ± 9.1	50F	Cancer	8 weeks; 7x/week	UINCARE and Xbox One Kinect; Follow the movement of the circles and sounds on the screen; 56 sets	Pain	
Qi et al. (2024)	62.8 ± 11.2	48M/30F	Cancer	8 weeks; 4x/week	mHealth app; VR-based Reactivity Training with Positive Thinking Intervention and Scene Interaction, 30 min	Anxiety; Depression	
Tsuda et al. (2016)	66.0 ± 4.0	10M/6F	Blood cancers	3 weeks; 5x/week	Nintendo Wii Fit; Hula Hoop and Basic Step; 20 min	Anxiety; Depression	
Villumsen et al. (2019)	67.6 ± 4.6	23M	Prostate cancer	12 weeks; 3x/week	Xbox 360 Kinect; Aerobic and strength exercise for 30 min using the Your Shape Fitness Evolved 2012, Sport and Adventure games	Fatigue	
Note:

M, Male; F, Female; VR, Virtual reality.

Quality assessment

The total scores of the PEDro scale in the included RCT articles exceeded five points, indicating high quality. In contrast, the overall quality rating of all the included non-RCT articles on the NIH scale was classified as “good” (Table 2). A total of eight articles were also classified as 1B level evidence, while four articles were categorised as 2B level evidence. Moreover, three, six, and three articles received A, B, and C levels of recommendation, respectively.

Table 2 Depiction of the quality and bias assessment.

PEDro scale		1	2	3	4	5	6	7	8	9	10	11	Total	LE	LR	
da Silva Alves et al. (2017)		N*	Y	N	Y	N	N	N	Y	Y	Y	Y	6/10	1B	B	
da Silva Alves et al. (2023)		Y*	Y	Y	Y	Y	N	N	Y	Y	Y	Y	8/10	1B	A	
Basha et al. (2022)		Y*	Y	Y	Y	N	N	N	N	Y	Y	Y	6/10	1B	B	
Feyzioğlu et al. (2020)		N*	Y	N	Y	N	N	Y	Y	Y	Y	Y	7/10	1B	B	
Hamari et al. (2019)		Y*	Y	Y	Y	N	Y	N	Y	Y	Y	Y	8/10	1B	A	
Abdelmoniem Ibrahim et al. (2024)		Y*	Y	Y	Y	Y	N	N	N	Y	N	Y	6/10	1B	B	
Park et al. (2023)		Y*	Y	Y	Y	Y	N	N	N	Y	Y	Y	7/10	1B	B	
Villumsen et al. (2019)		Y*	Y	N	Y	Y	N	Y	Y	Y	Y	Y	8/10	1B	A	
NIH scale	1	2	3	4	5	6	7	8	9	10	11	12	Total			
Qi et al. (2024)	Y	Y	Y	Y	Y	Y	N	N	N	Y	Y	NA*	8/11	2B	C	
House et al. (2016)	Y	Y	Y	Y	Y	N	N	N	Y	Y	Y	NA*	8/11	2B	C	
Mao, Chen & Wang (2024)	Y	Y	Y	N	Y	Y	Y	N	Y	Y	Y	NA*	9/11	2B	B	
Tsuda et al. (2016)	Y	Y	Y	N	Y	Y	N	N	N	Y	Y	NA	7/11	2B	C	
Notes:

Y, yes; N, no; NA, not applicable; NR, not reported.

* Not included in total score.

LE, level of evidence; LR, level of recommendation.

Quantitative synthesis

The meta-analysis outcomes implied a greater reduction in fatigue following the VRG intervention in comparison to COG (SMD, 0.77 [0.50, 1.04], p < 0.05, I2 = 61%) (Fig. 2A). The heterogeneity of these outcomes then indicated that subgroup analyses were necessary. Consequently, these analyses denoted that fatigue was substantially improved in intervention durations below 30 min (SMD, 0.44 [0.08, 0.81], p < 0.05, I2 = 22%) and above 30 min (SMD, 1.20 [0.78, 1.61], p < 0.05, I2 = 0%). Significant pain improvement was also observed in VRG than COG (SMD, −0.88 [−1.15, −0.60], p < 0.05, I2 = 36%) (Fig. 2B).

Figure 2 Forest plot illustrates the effects of VRG vs. COG intervention on fatigue (A) and pain (B).

Forest plot (A) used 30 min of exercise as the threshold for subgroup analysis. Studies: Hamari et al. (2019), Abdelmoniem Ibrahim et al. (2024), Villumsen et al. (2019), da Silva Alves et al. (2017, 2023), Basha et al. (2022), Feyzioğlu et al. (2020), Park et al. (2023).

In addition, the results showed that VRG intervention can significantly reduce anxiety (SMD, 0.66 [0.43, 0.89], p < 0.05, I2 = 84%) (Fig. 3A). Compared to patients aged between 55 and 70 years (SMD, 0.27 [−0.02, 0.56], p = 0.07, I2 = 0%), notable reduced anxiety levels was presented for VRG regarding cancer patients aged between 35 and 55 years (SMD, 1.31 [0.94, 1.68], p < 0.05, I2 = 0%). Interestingly, an insignificant difference changes in depression levels was also concluded (SMD, 0.65 [−0.08, 1.39], p = 0.08, I2 = 86%) (Fig. 3B).

Figure 3 Forest plot illustrates the effects of VRG intervention on anxiety (A) and depression (B) before and after intervention.

Forest plot (A) used age 55 years as the threshold for subgroup analysis. Studies: Abdelmoniem Ibrahim et al. (2024), Villumsen et al. (2019), Mao, Chen & Wang (2024), Qi et al. (2024), House et al. (2016), Tsuda et al. (2016).

Sensitivity analysis

By excluding samples one by one and switching effect model, it is found that no single study has an impact on the significance of the statistical results, and most indicators have the stability of calibration. However, due to the low p value (p = 0.08) and high heterogeneity (I2 = 81%) of the original model of depression indicators, the stability of the results after changing the model is weak.

Publication bias analysis

The analysis of publication bias revealed that the 12 articles met the minimum criteria for the funnel plot. Figures 4 and 5 depict a left-right symmetrical distribution, indicating a low likelihood of publication bias. In addition, for the four parameters of fatigue, pain, anxiety, and depression, Egger’s analyses revealed that the p values were 0.73, 0.69, 0.35, and 0.49, respectively, indicating no publication bias for all these parameters.

Figure 4 Funnel plot of publication bias for fatigue (A) and pain (B) in the VRG vs. COG intervention.

Figure 5 Funnel plot of publication bias for fatigue (A) and pain (B) in the VRG intervention.

Discussion

This meta-analysis evaluated the impact of VR exercise on pain, fatigue, depression, and anxiety among cancer patients. The findings indicated that VR exercise markedly enhanced pain and fatigue levels among cancer patients. Specifically, a more significant impact on fatigue improvement was noted when an intervention programme exceeding 30 min was utilised. Although the VR exercise also substantially improved anxiety levels among cancer patients aged between 35 and 55 years, no such improvement was observed in those aged between 55 and 70 years. This observation signified that VR exercise could be an effective non-pharmacological intervention for cancer patients.

It is worth noting that VR exercise has not shown any beneficial effects on depression. The analysis may be related to the small sample size and high heterogeneity. From a statistical perspective, the current p-value is close to 0.05, and this result should be viewed with caution. There are still different opinions on the impact of VR exercise on depression, and VR exercise may have certain potential to improve depression. One study (Seo et al., 2023) showed that overweight middle-aged women participated in a VR exercise program. The VR group used smart phones and head mounted displays for immersive bicycle training, while the control group used traditional indoor bicycles or received no intervention. After 8 weeks of intervention, the depression score of the VR group was significantly lower than that of the control group (F = 3.462, p < 0.001). The VR program also enhanced the fun and immersion of sports, which may contribute to sustained behavioral participation. Although the participants were inconsistent, they also supported my previous view.

Cancer patients frequently exhibit fatigue and typically report a diminished quality of life (de et al., 2018) during and following treatment (Savina & Zaydiner, 2019). Several studies have also indicated that chemotherapy or radiotherapy protocols are linked to various fatigue types (Ma, Li & Chan, 2023). Thus, VR games are an innovative therapeutic modality that generates highly interactive virtual scenarios (Wang et al., 2022). Players can then replicate specific exercise movements on the game screen by altering gestures and body movements of the upper and lower limbs, eliminating the necessity for complex control devices. This interaction introduces novel possibilities and enhances vitality within therapeutic applications (Xiang et al., 2021). Overall, this review demonstrated that VR exercise notably enhanced cancer patients’ fatigue. A subgroup analysis also suggested that intervention programmes exceeding 30 min appeared to yield superior outcomes. The selection of 30 min as the critical value is mainly based on the effectiveness and tolerability of exercise intervention, and exercise programs around 30 min have been widely studied and applied (Evans et al., 2016; Nouchi, Nouchi & Kawashima, 2020).

da Silva Alves et al. (2023) recruited 38 cancer patients from the oncology department of a hospital undergoing chemotherapy. These patients participated in a VR exercise intervention over a period of 7 weeks, with three sessions each week. The intervention utilised an Xbox 360 Kinect, focusing on gaming activities for the upper and lower extremities. Each intervention session lasted 47 min. The Functional Assessment of Chronic Illness Therapy-Fatigue Questionnaire was also employed to evaluate cancer-related fatigue. Consequently, the VR exercise group significantly reduced fatigue levels among cancer patients compared to the COG (intervention group = 14.79% vs. control group = −10.01%). This outcome aligned with the findings of this review.

The VR game intervention mechanism aimed at alleviating fatigue in cancer patients encompasses multi-system regulation. This process includes the activation of motor function via immersive physical interaction tasks, an increase in metabolic levels, and a reduction in the perception of exercise tolerance through enjoyable activities, disrupting the detrimental cycle of “fatigue-activity reduction”. Concurrently, sensory stimulation from VR can elevate cortisol and inhibit the release of pro-inflammatory cytokines (IL-6) by regulating the hypothalamic-pituitary-adrenal axis (Burrai et al., 2023). This process can reduce systemic inflammation. Moreover, the social interaction components and narrative design of VR can enhance psychological well-being, addressing anxiety and loneliness issues, increase self-efficacy, and indirectly mitigate fatigue through psychological-physiological interactions (Rutkowski et al., 2021). Nonetheless, the analysis of mechanisms in this study remained constrained by a small sample size, significant intervention heterogeneity, and inadequate data regarding long-term impacts. This outcome suggested that further investigation was required, incorporating neuroimaging and tailored programme design.

The management of cancer-related pain presents significant challenges (Thronæs et al., 2016). Notably, the characteristics and intricacies of chronic pain do not readily facilitate alterations in the perception of pain intensity (Marty et al., 2009). This review also indicated that the efficacy of VR exercise in alleviating cancer-related pain aligned with the outcomes of Abdelmoniem Ibrahim et al. (2024). The study involved 21 female cancer patients participating in an 8-week VR exercise programme featuring several activities (shooting balloons and recycling) conducted for 15 min three times weekly. Pain levels were also evaluated using a visual analogue scale (VAS), revealing that the VR exercise group experienced a significant reduction in pain levels. Typically, pain is a common symptom linked to breast cancer treatment. Hidding et al. (2014) recorded the incidence of post-operative shoulder and chest pain following breast cancer surgery. The incidence of surgery was reported at 75% and 82%, respectively. In addition, the risk of developing pain was established with level of evidence 1 in patients undergoing axillary lymph node dissection (ALND), radiation therapy, chemotherapy, or receiving therapies (Hidding et al., 2014).

Likewise, VR exercise interventions alleviated cancer-related pain via neural and psychological pathways. The immersive VR diverted attention from nociceptive signals, activating brain regions (anterior cingulate cortex) to attenuate pain processing. This finding was evidenced by functional near-infrared spectroscopy analysis demonstrating decreased activity in pain-related areas during VR engagement. A study (Birkhoff et al., 2021) contended a 1.4-point decrease in pain intensity (0–10 scale) following a 10-min VR session, with effects persisting for 24 h. This observation was highly attributed to the parasympathetic activation (high heart rate variability) and reduced cortisol levels. Goal-oriented VR tasks also enhanced self-efficacy while lowering anxiety, further influencing pain perception (Birkhoff et al., 2021). The mechanism by which VR exercise improves pain is uncertain, which may involve mechanisms such as distraction and cognitive reappraisal, activation of endogenous analgesic system, and improvement of emotional and psychological states. Pain perception is highly dependent on the brain’s attention allocation–when attention is focused on pain, the pain sensation will be amplified (Jin et al., 2016); The immersive experience of VR (such as gamified tasks and virtual scene interaction) can shift attention from the pain part to the target in the virtual environment (such as avoiding obstacles and completing action instructions), reducing the brain’s “priority processing” of pain signals. At the same time, VR can change the interpretation of pain through cognitive reappraisal (such as treating “pain” as a “task challenge”), reducing the subjective discomfort of pain. This process is related to the inhibitory effect of prefrontal cortex on pain related brain regions (such as insula and cingulate gyrus) (Li et al., 2011). Immersive experience or pleasant virtual scenes can promote the secretion of neurotransmitters such as endorphins and dopamine, which can directly inhibit the pain transmission pathway in the spinal dorsal horn and brainstem, and weaken the transmission of pain signals to the brain (Medina, Clarke & Hughes, 2024; Matamala-Gomez, Donegan & Świdrak, 2023). In addition, pain and negative emotions (anxiety, depression, helplessness) form a vicious cycle: emotional deterioration will enhance pain sensitivity, while pain exacerbation further aggravates emotional problems. Virtual scenes in VR exercise (such as natural landscapes and interactive games) can activate brain reward circuits (such as the nucleus accumbens), enhance pleasure, reduce the activity of amygdala (the core brain area dealing with negative emotions), and reduce the emotional component of pain. Completing motor tasks (such as rehabilitation training and physical challenge) in VR can improve the patient’s sense of control over the body, reduce the helplessness of “uncontrollable pain”, and indirectly reduce the pain experience (Matamala-Gomez, Donegan & Świdrak, 2023).

However, the practical obstacles to implementing VR exercise involve multiple challenges. Firstly, the existing content tends to be more entertainment oriented, lacking personalized and specialized design for different types of pain or rehabilitation needs, and the ability transfer effect between virtual scenes and real life is insufficient. Secondly, the lack of professional guidance may lead to safety risks, and there are also certain safety hazards for immunocompromised patients. In addition, elderly or low digital literacy groups have technological resistance and learning barriers, and long-term compliance is poor due to fading novelty or delayed effectiveness. High quality equipment has high maintenance costs, while data security and privacy risks also constrain the application of medical scenarios. These obstacles still need to rely on improving professional training and standard systems, strengthening policy support and cost control, in order to promote the effective implementation of VR exercise.

This study possessed several limitations. The analysis focused exclusively on the cancer population, neglecting other diseases. Furthermore, the sample included in this review was limited, comprising only 12 articles. Lower sample size and different variables such as age, disease type, and intervention program may lead to greater heterogeneity of individual indicators. Thus, subgroup analysis was conducted by analyzing variables such as age, disease type, and intervention program to try to explain the reasons for heterogeneity. Another issue was the limited quality of research on secondary indicators concerning the two primary assessment parameters in this review (RCT and single-arm trials) for assessing anxiety and depression parameters. In the results of depression indicators, high heterogeneity and close to the critical value of significance lead to poor stability of the results, so we should be cautious about the results of VR exercise on depression. Insufficient clarity regarding the mechanisms through which VR exercise enhances physical status in cancer patients was also observed. Hence, future studies should include more high-quality, multi-population RCT studies to strengthen the dissemination of results and to investigate the underlying mechanisms in greater detail. For clinical recommendations, middle-aged adults with cancer aged 35–55 years are advised to engage in VR exercise sessions lasting more than 30 min to improve their physical condition.

Conclusion

This review successfully indicated that VR exercise reduced fatigue, pain, and anxiety in cancer patients. Nevertheless, the effect of VR exercise on relieving depression is not clear, which may be a potentially effective non pharmacological intervention in this population.

Supplemental Information

Supplemental Information 1 PRISMA checklist.

Supplemental Information 2 Data.

Supplemental Information 3 Search strategy.

Supplemental Information 4 The audience.

Additional Information and Declarations

Competing Interests

The authors declare that they have no competing interests.

Author Contributions

Ruihan Wang conceived and designed the experiments, performed the experiments, prepared figures and/or tables, authored or reviewed drafts of the article, and approved the final draft.

Yunwei Wang conceived and designed the experiments, performed the experiments, prepared figures and/or tables, authored or reviewed drafts of the article, and approved the final draft.

Shiming Li analyzed the data, prepared figures and/or tables, authored or reviewed drafts of the article, and approved the final draft.

Shuoqi Li analyzed the data, prepared figures and/or tables, authored or reviewed drafts of the article, and approved the final draft.

Data Availability

The following information was supplied regarding data availability:

This is a systematic review/meta-analysis.

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
