# Peer review of "Effects of virtual reality exercise on fatigue, pain, and psychological status among cancer patients: a meta-analysis"

_PeerJ, doi:10.7717/peerj.20196_

## Round 0.1 · original submission · Major Revisions

I agree with the comments provided by R1 regarding the methodlogy (specifically the sensitivity analysis). Please address Reviewers' concers, I believe this manuscript has a great potentatial and addresses an important topic.

**PeerJ Staff Note**: Please ensure that all review, editorial, and staff comments are addressed in a response letter and that any edits or clarifications mentioned in the letter are also inserted into the revised manuscript where appropriate.

**PeerJ Staff Note**: It is PeerJ policy that additional references suggested during the peer-review process should only be included if the authors agree that they are relevant and useful.

**Language Note**: The review process has identified that the English language must be improved. PeerJ can provide language editing services - please contact us at [email protected] for pricing (be sure to provide your manuscript number and title). Alternatively, you should make your own arrangements to improve the language quality and provide details in your response letter. – PeerJ Staff

Reviewer 1 ·

Basic reporting

The manuscript is generally understandable but requires substantial language editing to meet publication standards. Many sections (e.g., Abstract, Methods, Discussion) are affected by awkward phrasing and grammatical issues. Consider a professional English-language editing service. The structure generally follows standard meta-analysis reporting guidelines (e.g., PRISMA), but some critical aspects are either unclear or inadequately presented, such as the detailed search strategy and specific definitions used in inclusion criteria. Figures and tables are relevant and mostly clear, but legends require more detail. For example, figure captions do not always explain the subgroups or statistical models used.

Experimental design

The research question is relevant and timely. However, key details about the methodology need to be revised or expanded:

- There is inconsistency in the reporting of inclusion/exclusion criteria (e.g., mixing RCTs and non-RCTs without clearly justifying this).
- The term "cohort articles" is unclear—please define what types of studies were included under this label.
The search strategy is not reported in sufficient detail. Please provide:
- Exact search strings.
- Inclusion of gray literature databases (if any).
- Justification for the 2015–2025 time frame.
The use of the PEDro and NIH tools is appropriate, but the scoring rationale should be clarified in-text. How were discrepancies resolved?

Validity of the findings

- Conversion methods (e.g., medians to means) must be justified more clearly. Hozo et al. is referenced, but application details are missing.
- Funnel plots are shown, but more robust publication bias tests (e.g., Egger’s test) are recommended due to the limited number of included studies.
- Subgroup analyses were performed, which is good practice. However, the justification for subgroup cutoffs (e.g., 30 minutes for duration) should be grounded in existing literature or theory.
- Consider sensitivity analyses to assess the robustness of findings.
- The conclusions overstate the findings slightly. For example, the benefit on anxiety was found with high heterogeneity (I² = 84%), which weakens the strength of that result.
- The authors state that depression was unaffected, but the evidence base for this conclusion (only four studies) is too small to be definitive.

Additional comments

The topic is highly relevant, particularly given the increasing interest in VR in rehabilitation. The inclusion of both physical and psychological outcomes is commendable.
- Inadequate reporting of methods, especially in search strategy and data extraction.
- Mixing RCT and non-RCT data without stratified analysis or clear justification.
- Insufficient discussion of limitations related to the heterogeneity and small sample sizes in subgroup analyses.
- Needs careful language revision throughout.

Reviewer 2 ·

Basic reporting

The manuscript is generally well-written, and the topic is highly relevant and addresses an important clinical need. Tables and figures are appropriately presented and support the findings. The inclusion of the PRISMA checklist demonstrates adherence to systematic review standards and is a strength of this article. The literature references are comprehensive and relevant.

However, certain elements need attention and improvement:
1. The interpretation of depression results requires clarification. The statement "VR exercise did not alleviate depression" is imprecise, considering the non-significant p-value (0.08) with a confidence interval that crosses zero.

2. The abstract and conclusions should more accurately reflect the findings, particularly those concerning depression.

Experimental design

The research question is well-defined and addresses a knowledge gap regarding VR interventions for cancer patients. The methodology is consistent with systematic review guidelines. The search strategy across four databases is comprehensive. Quality assessment using appropriate tools (PEDro scale, NIH scale) demonstrates methodological rigor.

However, certain elements need attention and improvement:
1. Lack of platform-specific analysis: The study combines different VR platforms (Xbox Kinect, Nintendo Wii, Pico Neo 4, etc.) without analyzing their differential effects on outcomes. Please explain the differences in their impact.

2. Although subgroup analyses by duration were performed for fatigue, the study lacks analysis by, for example, cancer type (breast, prostate, blood cancers, etc.), patient age groups, disease stage, or treatment modality. These factors may significantly affect the studied parameters.

3. The wide variety of VR programs, session duration (15-60 minutes), and frequency (1-7 times/week) creates substantial methodological diversity that limits the interpretability of their comparison.

Validity of the findings

1. The discussion lacks a detailed explanation of neurobiological mechanisms underlying VR's differential effects on various symptoms (pain vs. fatigue vs. anxiety)

2. There is no discussion of practical barriers to implementing such systems

3. Missing discussion of the costs and benefits of applying such technologies

4. Safety for immunocompromised patients should be discussed

5. Recommend proposing recommendations for standardizing clinical protocols

6. The non-significant depression results require more detailed discussion of potential causes.

Additional comments

This meta-analysis addresses an important clinical question and provides valuable evidence for the benefits of VR exercise in cancer patients. The methodology is generally sound, and the findings regarding fatigue and pain reduction are clinically significant. However, the study would benefit significantly from addressing intervention heterogeneity, expanding discussion of mechanisms, and providing more practical clinical guidance. With these revisions, this work could make a substantial contribution to evidence-based supportive care in oncology.

---

## Round 0.2 · Minor Revisions

I believe the manuscript itself is ready for publication but please, as one of the reviewers suggested, include the data that you have used as supplementary materials, as it facilitates replication, and enables the inclusion of your findings in future meta-analyses.

Reviewer 2 ·

Basic reporting

Thank you for enabling me to conduct a review of this revised manuscript. The authors have incorporated most of the suggestions proposed by the reviewer, primarily in the area of improving the metodological description. They havee added sensitivity analysis and improved the interpretation of results.
In the case of systematic reviews/meta-analyses, data should be readily available, e.g., as "supplementary material"

Experimental design

no comment

Validity of the findings

no comment

Additional comments

The authorss have made significant improvements to address the previous reviewer concerns.

---

## Round 0.3 · accepted · Accept

Dear Authors,

Thank you so much for uploading the data! I think you're ready for publication.